# Diagnostic Performance of an Artificial Intelligence Software for the Evaluation of Bone X-Ray Examinations Referred from the Emergency Department

**DOI:** 10.3390/diagnostics15040491

**Published:** 2025-02-18

**Authors:** Alejandro Díaz Moreno, Raquel Cano Alonso, Ana Fernández Alfonso, Ana Álvarez Vázquez, Javier Carrascoso Arranz, Julia López Alcolea, David García Castellanos, Lucía Sanabria Greciano, Manuel Recio Rodríguez, Cristina Andreu-Vázquez, Israel John Thuissard Vasallo, Vicente Martínez De Vega

**Affiliations:** 1Hospital Universitario QuironSalud Madrid, 28223 Madrid, Spain; rcanoalonso@gmail.com (R.C.A.); anuskafer83@hotmail.com (A.F.A.); anaalvarezvazquez@gmail.com (A.Á.V.); javcarrascoso@hotmail.com (J.C.A.); lopezalcolea@gmail.com (J.L.A.); dvdgarcia1995@gmail.com (D.G.C.); lucia.sanabria@quironsalud.es (L.S.G.); mrmrecio@gmail.com (M.R.R.); vmartinezdevega@malvaluz.com (V.M.D.V.); 2Department of Medicine Faculty of Medicine, Health and Sports Universidad Europea de Madrid, 28670 Madrid, Spain; 3Faculty of Biomedical and Health Science, Universidad Europea de Madrid, 28670 Madrid, Spain; cristina.andreu@universidadeuropea.es (C.A.-V.); israeljohn.thuissard@universidadeuropea.es (I.J.T.V.)

**Keywords:** artificial intelligence, musculoskeletal, radiography, X-ray, emergency, digital radiography, automatic, detection, diagnosis

## Abstract

**Background/Objectives**: The growing use of artificial intelligence (AI) in musculoskeletal radiographs presents significant potential to improve diagnostic accuracy and optimize clinical workflow. However, assessing its performance in clinical environments is essential for successful implementation. We hypothesized that our AI applied to urgent bone X-rays could detect fractures, joint dislocations, and effusion with high sensitivity (Sens) and specificity (Spec). The specific objectives of our study were as follows: 1. To determine the Sens and Spec rates of AI in detecting bone fractures, dislocations, and elbow joint effusion compared to the gold standard (GS). 2. To evaluate the concordance rate between AI and radiology residents (RR). 3. To compare the proportion of doubtful results identified by AI and the RR, and the rates confirmed by GS. **Methods**: We conducted an observational, double-blind, retrospective study on adult bone X-rays (BXRs) referred from the emergency department at our center between October and November 2022, with a final sample of 792 BXRs, categorized into three groups: large joints, small joints, and long-flat bones. Our AI system detects fractures, dislocations, and elbow effusions, providing results as positive, negative, or doubtful. We compared the diagnostic performance of AI and the RR against a senior radiologist (GS). **Results**: The study population’s median age was 48 years; 48.6% were male. Statistical analysis showed Sens = 90.6% and Spec = 98% for fracture detection by the RR, and 95.8% and 97.6% by AI. The RR achieved higher Sens (77.8%) and Spec (100%) for dislocation detection compared to AI. The Kappa coefficient between RR and AI was 0.797 for fractures in large joints, and concordance was considered acceptable for all other variables. We also analyzed doubtful cases and their confirmation by GS. Additionally, we analyzed findings not detected by AI, such as chronic fractures, arthropathy, focal lesions, and anatomical variants. **Conclusions**: This study assessed the impact of AI in a real-world clinical setting, comparing its performance with that of radiologists (both in training and senior). AI achieved high Sens, Spec, and AUC in bone fracture detection and showed strong concordance with the RR. In conclusion, AI has the potential to be a valuable screening tool, helping reduce missed diagnoses in clinical practice.

## 1. Introduction

The plain radiograph is the most widely used imaging technique globally and also the most frequently requested first-line test from emergency departments for identifying the most common osteoarticular pathologies, especially traumatic conditions such as fractures and dislocations [1].

Interpreting plain radiographs is a complex task that requires adequate training. In routine clinical practice, emergency physicians interpret these radiographs on their own, without awaiting the radiologist’s report. This can result in a significant number of diagnostic errors [2], with missed fractures being the most common. According to Guly HR’s study, missed fractures account for up to 80% of errors in the interpretation of emergency bone radiographs [3].

In response to this issue, several commercial companies have developed artificial intelligence (AI) software capable of detecting different items in plain radiographic images, yielding promising results. AI in plain radiography is here to stay, as it is a useful tool both for screening and decision-making, providing a second reading of the imaging test [4].

This study hypothesizes that AI, when applied to the interpretation of urgent osteoarticular radiographs, demonstrates high sensitivity and specificity, as well as good agreement with the interpretations made by radiology residents.

Therefore, the main objective of our study is to assess the sensitivity and specificity of the interpretations made by the AI software available in our department (Arterys MSK AI, Milvue, hereinafter Milvue) in comparison to the interpretations of senior radiologists (considered the gold standard) of the plain osteoarticular radiographs of patients attending the emergency department at Hospital Universitario QuironSalud Madrid. Additionally, this study aims to evaluate the level of agreement between the interpretation made by Milvue and those made by radiology residents in training.

## 2. Materials and Methods

### 2.1. Study Type and Subjects

We designed an observational, retrospective, descriptive, and double-blind study. The sample included all osteoarticular radiographs (pelvis, upper limb, and lower limb; excluding other anatomical locations, such as spine or ribs) performed on patients who visited the emergency department at Hospital Universitario Quironsalud Madrid between 15 October and 15 November 2022. The data were extracted from the hospital’s Picture Archiving and Communication System (PACS) and analyzed retrospectively. From the initial sample of 2487 eligible patients, the established inclusion and exclusion criteria were applied. The inclusion criteria consisted of selecting only patients over 18 years of age to avoid the potential interference of growth plate lines in fracture detection. The exclusion criteria included cases in which it was not possible to access the report of the radiograph interpretation performed by Milvue, which occurred in 25 patients, and cases in which radiographs were discarded due to poor diagnostic quality, affecting 3 patients, one of whom also lacked the Milvue report. After applying these criteria, the final sample was reduced to 792 patients (Figure 1).

### 2.2. Acquisition Protocol

The two conventional digital radiology devices available in our facilities were used (Discovery XR656 HD—GE Healthcare—and YSIO X.pree—Siemens Healthineers-).

### 2.3. Reading and Data Collection Protocol

The AI software (Arterys Chest MICA v29.4.0, developed by Arterys, a company that has been acquired by a Paris-based company called Milvue) was the world’s first online medical imaging platform fully cloud native, powered by AI and FDA cleared. It was a clinical application (CE marked as a Class IIa medical device) designed to process appendicular skeleton and pelvis radiographic series and identify three imaging findings (categorical variables): fracture, dislocation, and elbow joint effusion.

Each detected finding was localized in the image using a bounding box and was assigned a confidence label, either “positive” (continuous line) or “doubtful” (dashed line). Moreover, the algorithm provided a list of findings not detected in the current radiographic view (Figure 2).

All findings were detected by a deep learning model that processed all radiographic views included in the series. The AI results were integrated into the institutional PACS and the clinicians’ image viewer, displayed in a secondary capture apart from the original radiograph. In practice, radiologists first evaluated the bone X-ray images, and then reviewed the AI results on the reading workstation to assist in completing the radiology report. When the requesting physicians examined the bone X-ray in the viewer, they could also see the AI’s analysis.

According to the information provided by the commercial company, the deep learning algorithm within the AI musculoskeletal product consists of a convolutional neural network that detected the aforementioned findings in each radiographic view. The model was trained and validated using 1,262,467 and 157,181 radiographic images. Afterwards, it was calibrated on another independent database of 4759 images. The data used for model development were sourced from a multicenter database, which included both pediatric and adult patients). However, our research team was not involved in the development or programming of the AI software. Our role was solely as end users, acting as independent testers of a commercially available product, without any direct commercial relationship with the company.

Each of the radiographs in our study was read and interpreted by the AI software (Milvue) as well as by the resident and the senior attending radiologist (gold standard). Only if the senior radiologist’s reading revealed any doubtful findings was a consensus sought with the reading of another senior radiologist. In all cases, neither the residents nor the attendings had access to the clinical data of the emergency episode through the hospital’s Electronic Health Record.

A first-year radiology resident was responsible for reading the radiographs, recording all study variables and a series of additional items in a table, without access to the report provided by Milvue for each of them. Their reading could result in a positive, negative, or doubtful outcome for each of the items listed in the “variables” section.

Subsequently, a senior attending radiologist with 13 years of experience indeendently reviewed these radiographs, also without access to the Milvue report. The reading performed by the senior attending radiologist was considered the gold Sstandard; therefore, it could only have a positive or negative outcome for the findings in question (the “doubtful” category could not be applied).

Finally, a second radiology resident independently collected the information provided by Milvue for each of these radiographs. This allowed for a subsequent comparison of the results obtained by the AI, the resident, and the senior radiologist, as well as an assessment of the level of agreement between the readings of the AI and the radiology resident (Figure 3).

### 2.4. Study Variables

A series of sociodemographic and radiological variables were analyzed for each patient.

Regarding sociodemographic variables, age and sex of the patients were included; also considering the number of radiographic projections performed, radiograph quality, joint groups (small and large joints, as well as flat/long bones), and the studied joint.

The radiological variables that Milvue is trained to recognize and that we included in the study were as follows:-Fracture.-Dislocation.-Joint effusion (only available for elbow radiographs).

In addition, we have studied other variables that Milvue is not trained to recognize. We believe that analyzing these variables is highly relevant for evaluating the overall impact of Milvue in a way that aligns with daily clinical practice, as well as its strengths and weaknesses compared to the systematic interpretation of radiographs performed by radiology residents and senior radiologists. These variables are as follows:-Sequelae of a fracture; understanding this variable as a chronic fracture or dislocation.-Arthropathy, which includes osteoarthritis and erosive arthritis; however, in our sample, we did not have cases of erosive arthritis, so in our study, arthropathy refers exclusively to osteoarthritis.-Focal lesion.-Anatomical variant.-Other findings: includes other findings not covered in previous categories.

Our study was approved by the Ethics Committee of our institutional review board (Hospital Fundación Jiménez Díaz; Grupo QuironSalud) on 10 January 2023 (no 01/23) and reapproved after minor revisions on 28 February 2023 (no 04/23) with the code EO017-23_HUQM.

## 3. Results

### 3.1. Demographic Characteristics

We briefly summarize the distribution of the demographic characteristics of our sample, which is detailed in Table 1.

In our series, the median age of the patients was 48 years, with 51.4% women (*n* = 407). The majority of radiographs were of optimal quality (97.2%) and had two projections per study (90.2%). Dividing the anatomical regions into three major families was considered useful for a better evaluation of diagnostic capabilities and the degree of agreement between Milvue and the residents. The groups were organized as follows:**Small joints** which included hand, feet, fingers, calcaneus, wrist, and ankle. This was the most frequently found group, accounting for up to 51.4% of the cases.**Large joints** which included shoulder, elbow, pelvis or hip, and knee. This was the second most frequent group, representing 43.3% of the cases.**Flat or long bones** which included clavicle, humerus, forearm, femur, and tibia. This was the least frequent group (5.3%).

### 3.2. Prevalence

As can be seen in Table 2, from the reading of the gold standard, we can estimate the natural prevalence of each diagnosis in the studied sample. Acute fractures stand out due to their frequency (overall prevalence of 16.9%), with the highest numbers found in the small joint group (20.6%). For the other variables evaluated by Milvue, their overall prevalences were 2.5% for dislocations and 25% for joint effusion in elbow radiographs.

Regarding the variables not analyzed by Milvue, the prevalence of 24.6% for the “other findings” variable stands out. Despite this, it proved to be a very unresolving item because the findings included in this category were recorded under a free-text criterion. Several of them could be observed simultaneously in the same patient and were sometimes of little clinical relevance in the emergency setting. Some examples include the increase in periarticular soft tissues, enthesopathy, or the presence of post-surgical material.

Also notable is the prevalence of the variables’ arthropathy and anatomical variants, at 19.8% and 12.6%, respectively. Regarding the first variable, most of the examinations corresponded to osteoarthritic changes in elderly patients, as would be expected given the age distribution in the study sample. On the other hand, the most prevalent anatomical variant was the fabella in knee radiographs, followed by other accessory bones in ankle and foot radiographs.

The low prevalence of focal bone lesions (1.9%) can be explained by the fact that the sample was extracted from studies requested by the emergency department, with most being benign bone lesions diagnosed incidentally, except for one pathological fracture in the femur secondary to a bone metastasis from a lung carcinoma.

### 3.3. Analysis of the Resident’s and AI’s Performance Validity Compared to the Gold Standard

#### 3.3.1. Acute Fracture

The analysis of the accuracy of the resident’s and Milvue’s readings compared to the gold standard can be seen in Table 3, Table 4 and Table 5. Additionally, we would like to highlight the analysis of the results marked as doubtful by AI and the resident and how many of them were confirmed by the gold standard. Therefore, doubtful cases were excluded in the calculation of sensitivity, specificity, PPV, NPV, and ROC area.

Regarding fracture detection (Table 3), both readings present generally very high values for sensitivity, specificity, positive predictive value (PPV), negative predictive value (NPV), and area under the curve (AUC). In the overall cases, Milvue achieved a sensitivity of 95.8% (CI 90.5–98.6), a specificity of 97.6% (CI 96–98.6), and an AUC of 0.967 (0.948–0.986).

If we pay attention to the subgroup analysis, the trend is similar, especially highlighting the data of the resident in the subgroup of flat/long bones, where sensitivity reached 100% (CI 75.3–100) and specificity reached 100% (87.7–100).

Furthermore, it is worth noting that Milvue generated more doubtful results than the resident (58 doubtful cases from the AI vs. 12 from the resident), with the positivity for those doubts being lower than in the resident’s reading (only 25.9% of the doubtful cases in the AI report were truly positive for fractures, while 50% of the doubtful results from the resident’s reading were ultimately fractures).

#### 3.3.2. Acute Joint Dislocation

As we mentioned earlier, the prevalence of dislocations was very low, which means that their sensitivity values are only valid in the group of large joints in the resident’s reading, although with very wide confidence intervals. Thus, we found a high sensitivity of 77.8 (CI 52.4–93.6) in the overall cases for the resident’s reading, which improved to 84.6 (CI 54.6–98.1) in the large joint group. However, AI obtained worse sensitivity results of 35 (CI 15.4–59.2) and 35.7 (CI 12.8–64.9) for the overall cases and for the large joint group, respectively (Table 4).

On the other hand, the specificity values were excellent for all readings and all groups, ranging between 99.5 and 100. Additionally, we highlight that the resident achieved excellent AUC values in all subgroups, especially in the large joint group, with an AUC of 0.923 (CI 0.821–1); whereas the AI only achieved a good AUC in the small joint subgroup, with an AUC of 0.831 (CI 0.504–1).

The specific case of acromioclavicular dislocations is particularly striking, as none of these cases were read as positive or doubtful by Milvue, making it impossible to perform the corresponding statistical calculations.

#### 3.3.3. Elbow Joint Effusion

Our sample included twenty-eight elbow X-rays, of which only seven showed joint effusion. These prevalent data influenced the calculation of the other variables. Thus, despite finding a sensitivity of 100 (CI 54.1–100) in the Milvue reading and 100 (CI 59–100) in the resident’s reading, the 95% confidence intervals were very wide (Table 5).

In contrast, specificity and negative predictive values were more reliable for both readings, as well as their confidence intervals, with excellent areas under the curve. Notably, for the AI reading, a specificity of 94.4 (CI 72.7–99.9) was achieved, with an AUC of 0.972 (CI 0.918–1).

The AI recorded four doubtful cases of joint effusion, of which only one was confirmed by the gold standard. The resident did not record any doubtful cases.

#### 3.3.4. Degree of Agreement Between the Resident and AI

Finally, we also evaluated the degree of agreement between the two readers using the Kappa coefficient. Generally speaking, a Kappa value above 0.4 indicates moderate agreement, above 0.6 is substantial, and above 0.8 is considered almost perfect.

Figure 4 is a heat map. Regarding fractures, in the overall group, the Kappa value was 0.69, and it was particularly high in the large joint group (0.79). In terms of joint dislocations, moderate agreement was found in both the overall group and the large joint group, with Kappa values around 0.4.

In the small joint group, the agreement was only fair, with a Kappa of 0.3. The flat and long bones group could not be assessed because Milvue did not identify any dislocations in this group. Regarding elbow joint effusion, the agreement was substantial, with a Kappa value of 0.719.

However, it is important to note that Kappa values may depend on the prevalence of the evaluated condition, and the prevalence of fractures was significantly higher than that of dislocations or elbow joint effusion.

### 3.4. Analysis of the “Other Findings” That Milvue Has Not Been Trained to Detect

Below, we carry out a brief analysis of the variables that our AI software has not been trained to detect and that are reflected in Table 6. For the calculation of these variables, we only took into account the reading by the resident and the gold standard.

The low prevalence of most variables only makes the analysis of two of them interesting: anatomical variants and arthropathy. In the anatomical variant variable, sensitivity reaches 90% in the large joint group, and specificity and NPV values were higher than the PPV values, with particularly high AUCs for the large joint group. Regarding the detection of the arthropathy variable, sensitivity stands out in the large joints at 79.8%, with high specificity, NPV, and AUC values ranging between 0.74 and 0.891.

## 4. Discussion

Of all the applications that use AI in the field of Medicine, a large part of them focuses specifically on radiology [5,6], with the number of related publications having grown exponentially in recent years [7], especially those applied in emergency radiology [8,9,10].

The incorporation of AI in musculoskeletal radiology is bringing a significant change in the diagnosis and analysis of pathologies [11]. Its use allows the automatic detection of fractures [12,13], dislocations [14,15], and bone lesions with great precision [16]. Likewise, it facilitates the detection of joint effusions [17,18], the estimation of bone age using the Greulich and Pyle method [19,20], and the measurement of angles in bone structures such as the Cobb angle for scoliosis [21,22].

It is also capable of generating three-dimensional reconstructions that support surgical planning or diagnosing and monitoring the progression of metabolic diseases such as osteoporosis [23,24].

Many of these applications are of particular relevance in work environments where the radiologist works remotely, in settings with a high demand for radiological examinations (such as in the emergency department), or in both circumstances simultaneously. In these cases, the number of examinations reported by the radiologist is on demand, or the X-rays are directly evaluated by emergency physicians.

In this field, it has been demonstrated that the use of AI increases the level of accuracy in diagnoses made using plain radiography, allowing both general practitioners and radiology residents to enhance their ability to detect various items [25,26]. Thus, according to the study by Oppenheimer et al., the use of AI in the interpretation of X-rays by radiology residents could increase the sensitivity for fracture detection by up to 7% [27].

In the case of plain musculoskeletal radiography, the items AI has been trained to detect may vary depending on the provider’s design, but in general, they have been validated for the detection of acute fractures. There are numerous bibliographic references that have evaluated its potential, both in adults and in pediatric populations. In adults, Duron et al. have described that AI increases the diagnostic sensitivity of physicians and reduces the number of false positives, without resulting in an increased reading time. Furthermore, they report better results for AI compared to humans, although the design of their study presents an inclusion bias, as the most obvious fractures were excluded from the sample [1]. Similarly, in the study by Wood, they found similar findings, demonstrating that the use of AI by radiologists allowed them to improve fracture detection, in addition to performing at a faster pace and with less uncertainty in their diagnosis [28].

In the same way, and including both adult and pediatric populations, Regnard et al. describe in their study that AI is capable of increasing the detection of fractures that may go unnoticed by radiologists, so that the combination of AI and radiologists results in the best diagnostic approach [14].

In our study, we aim to reflect routine clinical practice, not only evaluating the accuracy of Milvue but also comparing the AI results with those of a radiology resident. This comparison is important, as the interpretation of a trauma specialist can differ significantly from that of emergency department clinicians, which may complicate direct comparisons. To make the evaluation more relevant, we chose to use the radiology resident’s readings as a closer approximation to the interpretations of emergency department clinicians.

Regarding **fracture detection** (Table 3), a particularly notable finding was that the readings by Milvue and the resident show generally very high values for Sens, Spec, PPV, NPV, and AUC, with data similar to those described in the literature. (Figure 5 and Figure 6).

In the overall cases, Milvue achieved a sensitivity of 95.8, a specificity of 97.6, and an AUC of 0.967. These results are consistent with those described in the literature, such as in the study by Franco et al., where AI achieved a sensitivity for fracture detection of 91.3 (CI 87.6–94.3) and a specificity slightly lower than that obtained in our study, at 76.7 (CI 71.5–81.3) [29]. Similarly, in the study by Xie et al., they obtained a high sensitivity that varied by location between 0.83 and 0.91, with an AUC greater than 0.92 for all studied locations [30].

What we found particularly noteworthy was that in the overall group, we recorded 12 doubts by the resident, and 50% of them were confirmed by the gold standard. Milvue provided a higher number of doubtful cases—58 cases—but the proportion of positive cases was lower (only 25.9%) (Figure 7).

This difference in the positivity of cases marked as doubtful was influenced by the presence of anatomical variants in the following:Ankle and foot: On six occasions, Milvue marked the fracture variable as doubtful in cases with a bipartite medial sesamoid (two patients), an accessory sesamoid at the base of the 5th metatarsal, synphalangism, os peroneum, and os naviculare. (Figure 8).Hand: Milvue marked the fracture variable as doubtful in the case of multiple accessory ossicles.Wrist: On four occasions, Milvue marked the fracture variable as doubtful in cases of os paranaviculare, os trapezium secundarium, os ulnar styloid, and os paratrapezium. However, Milvue did not detect fractures in three cases of os ulnar styloid, two cases of accessory ulnar styloid, nor in cases of os hypolunatum and os epilunatum.

Overall, the presence of anatomical variants did not cause any diagnostic confusion for the resident, except in two patients with os paranaviculare and os paratrapezium, which were mistakenly classified as fractures.

Regarding the detection of **dislocations** (Table 4), we observed better sensitivity results in the resident’s reading compared to that of the AI, with a high sensitivity of 77.8% in the overall cases, which improved to 84.6% in the large joint group. In contrast, the AI showed lower sensitivity values, with 35% in the overall cases and 35.7% in the large joints. However, specificity values were excellent for all readings and groups, reaching between 99.5% and 100%. (Figure 9 and Figure 10).

All these results were heavily influenced by the low prevalence of dislocations in our sample. If we refer to the literature, in the study by Regnard et al., they obtained sensitivity and specificity values for AI readings that were higher than those shown in our study, with 89.9 and 99.1, respectively [14]. However, they also align with our research in that the sensitivity and specificity values are higher in the radiologist’s reading compared to AI.

Finally, the specific case of acromioclavicular dislocations is particularly striking, as none of the cases were reported as positive or doubtful by Milvue (Figure 11). We have also not found studies that evaluate the diagnostic capacity of AI in detecting acromioclavicular dislocations. This leads us to propose that future updates of the AI system should focus on expanding the training dataset to include a larger sample of acromioclavicular joint cases, particularly those with pathological conditions, in order to improve the system’s ability to accurately detect these dislocations.

Given our interest in integrating AI tools into clinical practice, after concluding the study, we had the opportunity to meet with Milvue representatives on May 11, 2023. During this meeting, we presented our study findings, including identified pitfalls and areas for improvement. The company welcomed our feedback and committed to considering user input worldwide to refine and enhance the software in future updates, aiming to improve its diagnostic performance.

Regarding the detection of **joint effusion in the elbow** (Table 5), the low prevalence of joint effusion in our sample affected the calculation of the variables. Although a sensitivity of 100% was found in both the Milvue and the resident’s readings, the confidence intervals were very wide. On the other hand, specificity and negative predictive values were more reliable for both readings, with the AI reading standing out in particular, showing a specificity of 94.4% (Figure 12).

The data obtained in our study are consistent with those presented in the study by Huhtanen [17], in which, after analyzing more than 215 lateral elbow X-rays using AI and comparing them with three radiologists, they obtained sensitivity, specificity, and AUC values for the AI detection of joint effusion in the adult group of 91.7 (CI 87–90.1), 92.4 (CI 89.2–95.6), and 0.966 (CI 0.96–0.97), respectively.

However, in their study, Dupuis et al. [31] analyzed 1637 pediatric emergency elbow X-rays with AI, obtaining a sensitivity >89% for the detection of fracture and joint effusion, similar to what has been described in the literature. Nevertheless, the specificities were lower than those reported in previous studies, with 63% for joint effusion and 77% for fractures. They also found NPV >92% and PPV ranging between 54 and 73%. These results may have been influenced by the age group of the sample.

After analyzing the results of our study and those found in the literature, we can conclude that both the AI system and the resident demonstrated NPVs greater than 95% and AUC values above 0.8, with the exception of joint dislocations, likely due to their low prevalence in our sample. These results were accompanied by high 95% CIs, indicating statistically reliable performance. This suggests that **AI has the potential to serve as an effective screening tool in emergency departments**, efficiently identifying normal musculoskeletal radiographs and integrating smoothly into the workflow. By enhancing clinicians’ autonomy, AI can help reduce the workload of radiologists, allowing them to focus on more complex cases. However, since many of the doubtful cases classified by AI were not confirmed as true positives, emergency physicians should continue to consult radiologists for these specific cases. While AI can assist in the diagnostic process, it cannot yet fully replace the human touch in patient care, as the expertise and judgment of clinicians remain essential, particularly in uncertain or complex cases.

Finally, we analyzed the **degree of agreement between the readings of the AI and the radiology resident** (which is comparable to that of an emergency physician), a particularly innovative aspect that we have not found analyzed in the literature. This degree of agreement was assessed by calculating the Kappa coefficient, represented by a heat map in Figure 4.

For the overall sample, in the detection of fractures, the Kappa coefficient resulted in 0.69 (moderate agreement), and it was particularly high for the subgroup of large joints, with a result of 0.79 (substantial/almost perfect agreement).

A moderate correlation was also noteworthy, with a Kappa coefficient of 0.71. The correlation ranged from acceptable to slight in the detection of dislocations.

Despite our study design efforts, we must acknowledge several **limitations**. First, it was not a multicenter study, and the population analyzed was limited to adults. Additionally, the gold standard used in our study was the interpretation of a senior radiologist, with no other confirmatory tests considered.

Another limitation stems from the low prevalence of certain conditions in our sample, which may impact the results, particularly in statistical parameters such as ROC areas, confidence intervals, and Kappa values.

Furthermore, the readings of emergency department physicians were not included. In our effort to reflect routine clinical practice, we aimed not only to evaluate the accuracy of Milvue but also to compare its results with those of a radiology resident, as their readings were considered a closer approximation to those of experienced emergency department clinicians. However, further research is needed to determine whether AI can achieve a similar level of accuracy in musculoskeletal radiographs as emergency department physicians.

An additional important limitation is the lack of available clinical information, which is crucial in the emergency room setting. Without this context, it becomes difficult to properly localize lesions, as attention is not focused on the areas of clinical interest that are unknown to us.

Finally, our study was limited to the detection of joint effusion in the elbow joint because the AI model was not trained to identify it in other joints. We hypothesize that, as AI technology continues to develop, future studies that include additional joints may improve the detection of joint effusion.

## 5. Conclusions

Our study demonstrates that AI applied to the interpretation of standard X-ray examinations shows promising performance in detecting fractures in large joints, positioning it as a useful tool for screening in emergency departments. However, the AI produced more “doubtful” results than the resident, with a lower proportion of positive findings, and showed limitations in detecting dislocations, particularly in the acromioclavicular joint.

Interobserver agreement between the AI and the resident ranged from moderate to substantial across all variables, highlighting its high reliability. These findings underscore the potential of AI to complement medical diagnosis, helping to reduce the rate of undetected injuries and optimize screening in high-demand clinical settings.

## Figures and Tables

**Figure 1 diagnostics-15-00491-f001:**
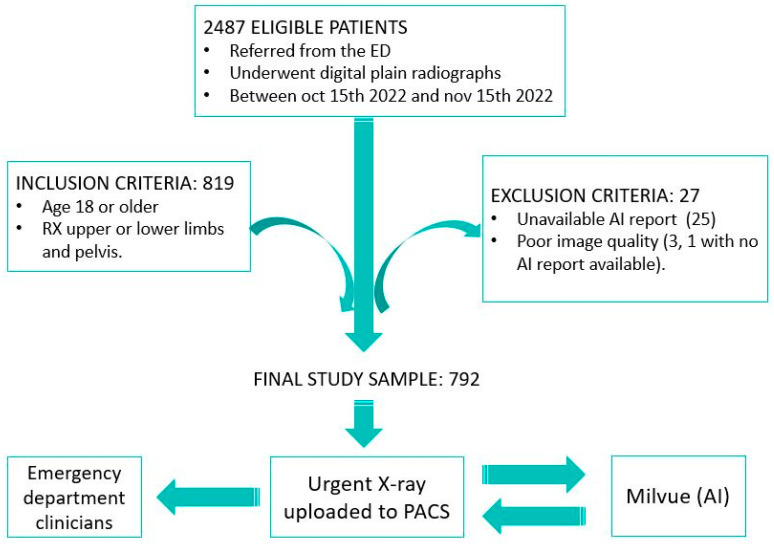
Overview of the patient selection process for the study. After applying inclusion and exclusion criteria, 792 patients with X-rays of the appendicular skeleton and pelvis were included in the final sample. Musculoskeletal radiographs uploaded to the PACS were automatically analyzed by AI, which generates a separate report to be displayed alongside the original X-ray so that both radiologists and clinicians can review it.

**Figure 2 diagnostics-15-00491-f002:**
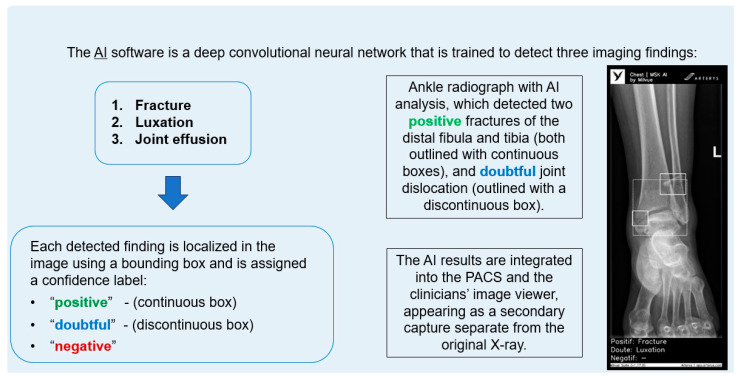
Diagram summarizing the functionality of the AI software and an example analyzing an ankle X-ray.

**Figure 3 diagnostics-15-00491-f003:**
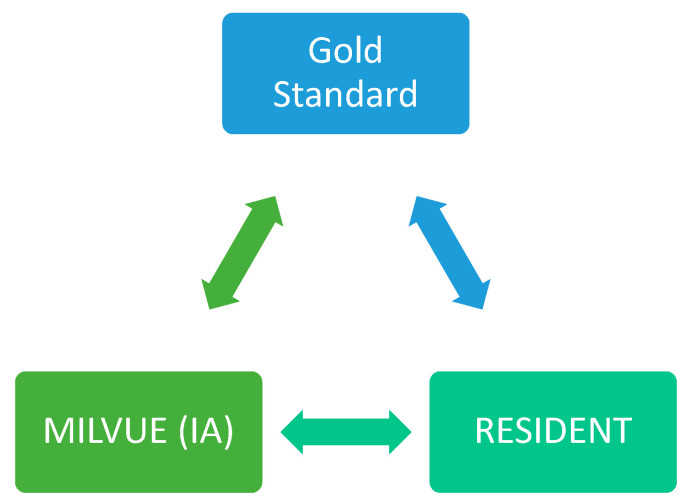
Graph showing the three readers of the X-rays. (GS: gold standard).

**Figure 4 diagnostics-15-00491-f004:**
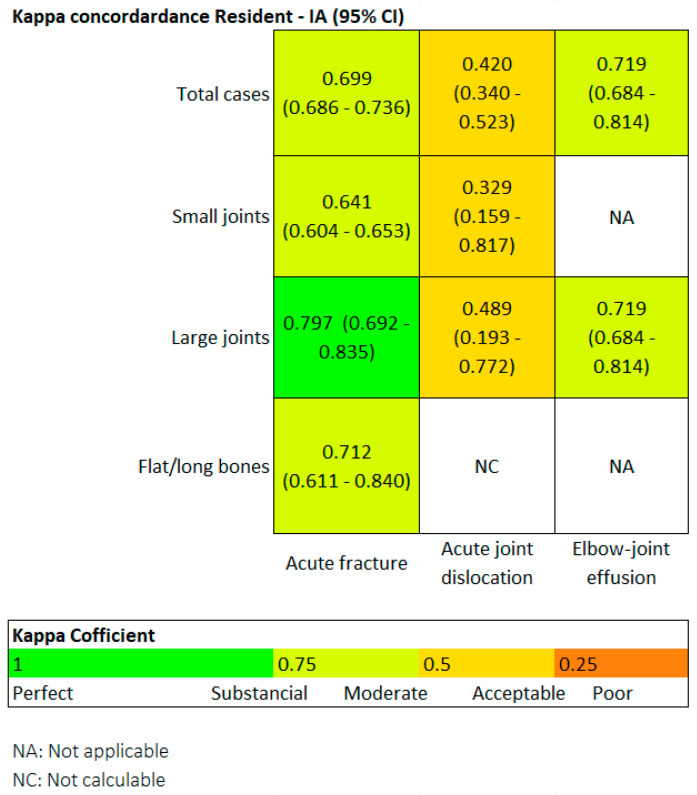
Heat map. For fractures in the overall group, Kappa was 0.69, and it was particularly high for the large joint group (0.79).

**Figure 5 diagnostics-15-00491-f005:**
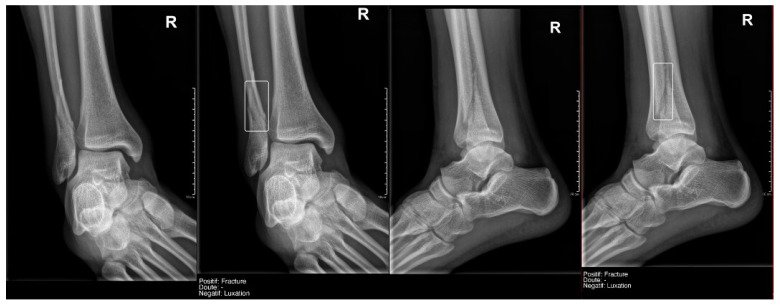
True positive fracture of the distal fibula correctly recorded by AI and radiology resident.

**Figure 6 diagnostics-15-00491-f006:**
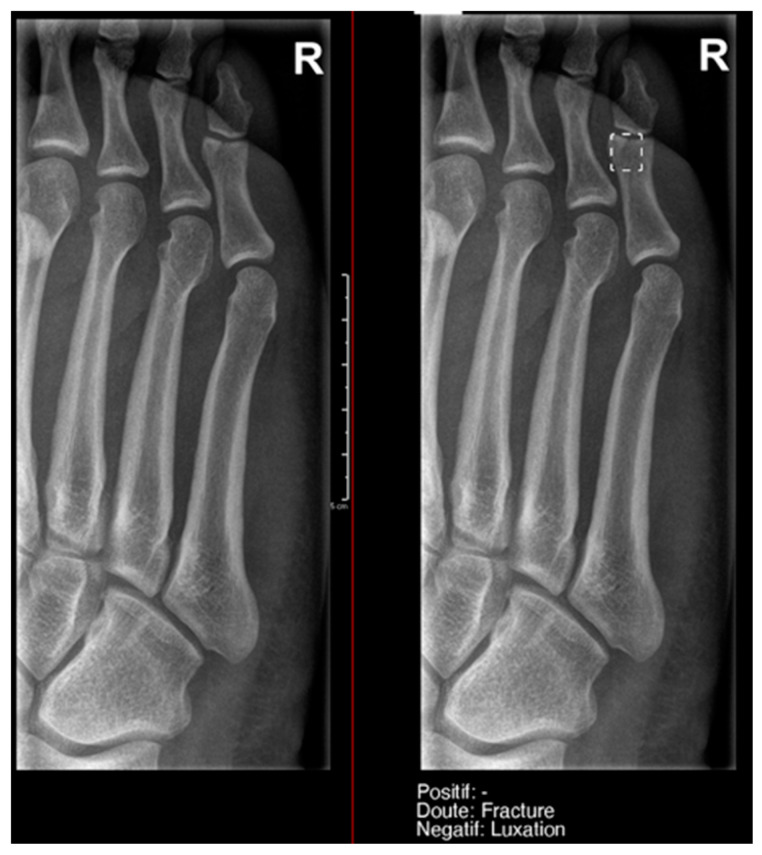
Fracture of the proximal phalanx of the fifth finger. It was recorded as doubtful by AI, but as positive by the radiology resident.

**Figure 7 diagnostics-15-00491-f007:**
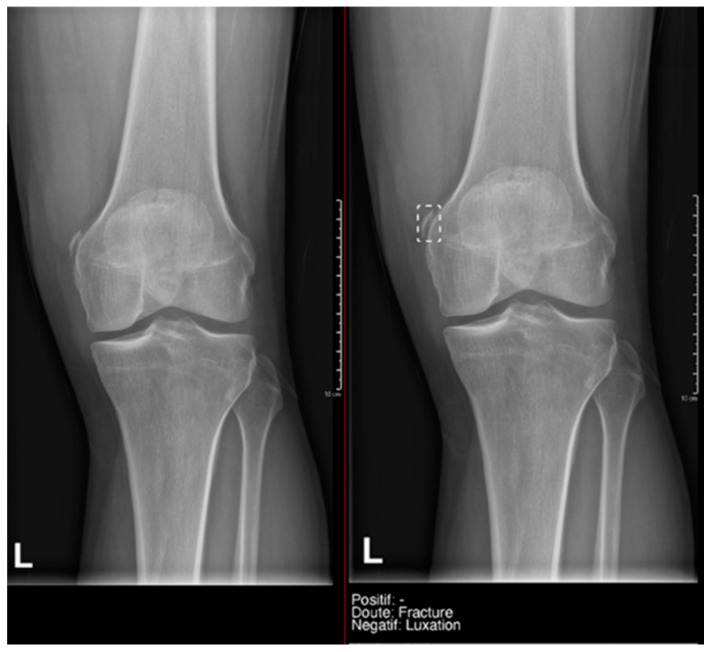
Example of a doubtful fracture marked with a dashed-line box on the AP knee radiograph, which corresponded to a Pellegrini–Stieda lesion.

**Figure 8 diagnostics-15-00491-f008:**
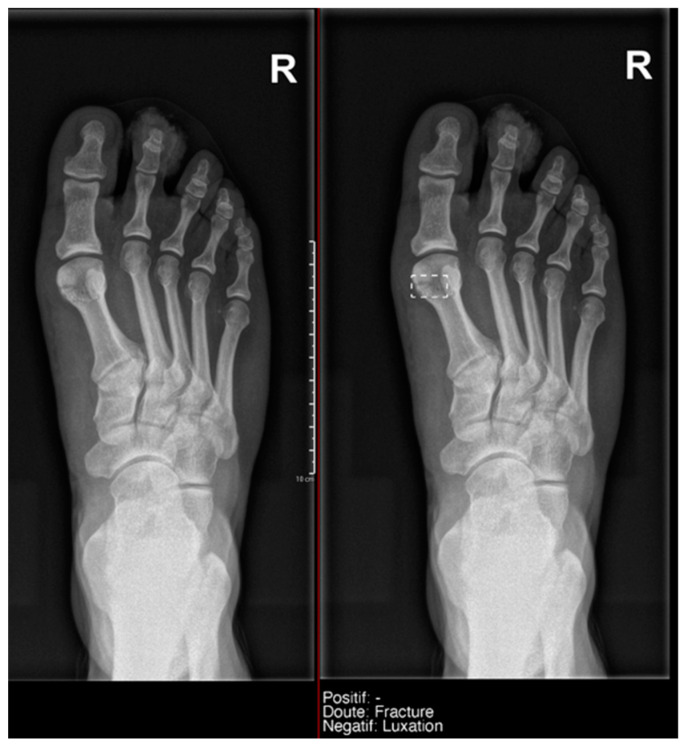
Example of an anatomical variant (bipartite hallux sesamoid), which was recorded as doubt fracture by AI and as negative by the radiology resident.

**Figure 9 diagnostics-15-00491-f009:**
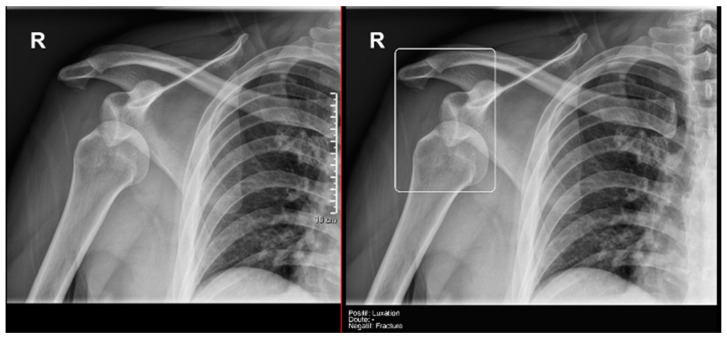
Glenohumeral joint dislocation that was correctly recorded by both the AI and radiology resident.

**Figure 10 diagnostics-15-00491-f010:**
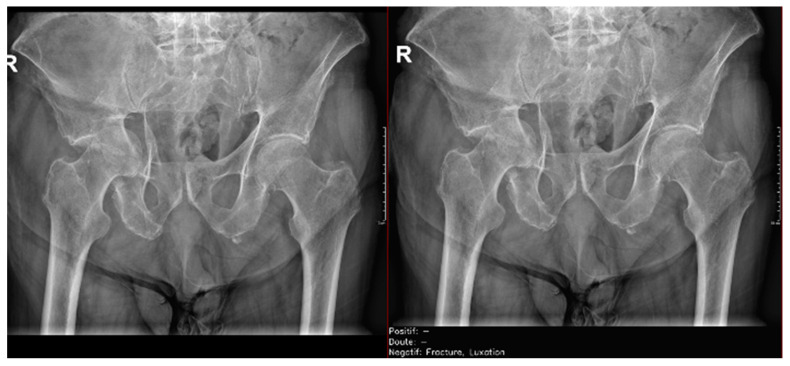
Pelvic fracture and coxofemoral dislocation, which was recorded as negative by AI but correctly detected by the radiology resident.

**Figure 11 diagnostics-15-00491-f011:**
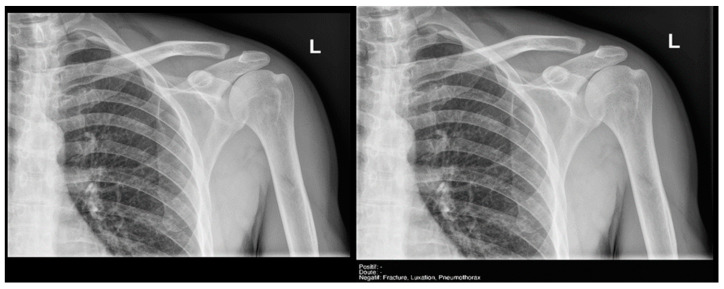
Acromioclavicular joint dislocation recorded as negative by AI and positive by radiology resident.

**Figure 12 diagnostics-15-00491-f012:**
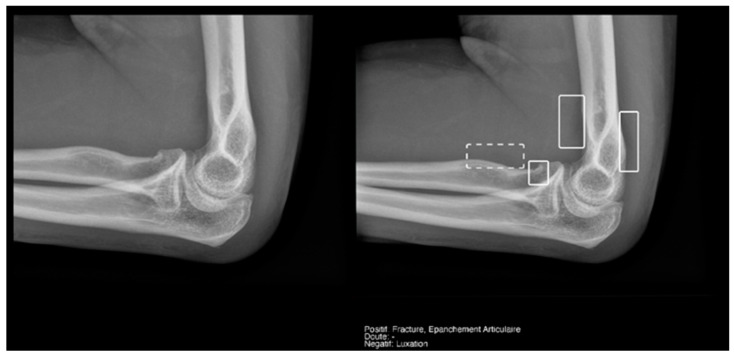
Radial head fracture and joint effusion, correctly detected by AI and radiology resident.

**Table 1 diagnostics-15-00491-t001:** Study sample characteristics.

		Cases
		(*n* = 792)
**Patient’s age (years old) median [Q1; Q3] ***		48.0 [33.0; 61.5]
**Gender (*n*,%)**		
	Men	385 (48.6)
	Women	407 (51.4)
**Radiological projections (*n*,%)**		
	1	62 (7.8)
	2	714 (90.2)
	3	12 (1.5)
	4	4 (0.5)
**X-ray image quality (*n*,%)**		
	Optimal	774 (97.8)
	Average	18 (2.2)
**Joint groups (*n*,%)**		
	Small joint	407 (51.4)
	Large joint	343 (43.3)
	Flat/long bones	42 (5.3)
**Breakdown by joints (*n*,%)**		
	Knee	151 (19.1)
	Ankle	127 (16.0)
	Shoulder	99 (12.5)
	Foot	77 (9.7)
	Wrist	68 (8.6)
	Pelvis–Hip	65 (8.2)
	Hand	53 (6.7)
	Fingers	47 (5.9)
	Toes	31 (3.9)
	Elbow	28 (3.5)
	Tibia	11 (1.4)
	Forearm	10 (1.3)
	Humerous	9 (1.1)
	Clavicle	6 (0.8)
	Femur	6 (0.8)
	Calcaneus	4 (0.5)

* Q1: first quartile = percentile 25. Q3: third quartile = percentile 75.

**Table 2 diagnostics-15-00491-t002:** Prevalence.

		Overall Cases	Large Joints	Small Joints	Flat/Long Bones
		(*n* = 792)	(*n* = 343)	(*n* = 407)	(*n* = 42)
**Prevalence *** **(*n*, % [IC 95%])**					
	Acute fracture	134 (16.9 [14.0–19.2])	37 (10.8 [7.1–13.8])	84 (20.6 [16.0–24.5])	13 (30.9 [18–48.1])
	Acute joint dislocation	20 (2.5 [1.4–3.6])	14 (4.1 [2–6.4])	3 (0.7 [0.15–2.1])	3 (7.1 [0.6–16.5])
	Chronic fractures	25 (3.2 [2–4.5])	9 (2.6 [1–4.6])	14 (3.4 [1.9–5.7])	2 (4.8 [0.6–16.2])
	Arthropathy	157 (19.8 [17–22.8])	99 (28.9 [24–34])	49 (12.4 [9–15.6])	9 (21.3 [10–36.8])
	Focal lesion	15 (1.9 [1.1–3.1])	9 (2.6 [0.9–4.3])	5 (1.2 [0.4–2.8])	1 (2.4 [0–7.2])
	Anatomical variant	100 (12.6 [10–15.2])	40 (11.7 [8.5–15.6])	60 (16.8 [11–18.6])	0 (0)
	Elbow joint effusion (*N* = 28)	7 (25.0 [11–44.9])			
	Other findings	195 (24.6 [21–27.6])	109 (31.8 [27–36.8])	77 (18.9 [15–22.9])	9 (21.4 [10–36.8])

* According to gold standard reading. CI: 95% confidence interval.

**Table 3 diagnostics-15-00491-t003:** Acute fracture.

		Overall Cases		Large Joints		Small Joints		Flat/Long Bones	
		(*n* = 792)		(*n* = 343)		(*n* = 407)		(*n* = 42)	
		Radiology Resident	IA Software	Radiology Resident	IA Software	Radiology Resident	IA Software	Radiology Resident	IA Software
**Acute fracture (ratio of doubtful cases */certain)**		12/780	58/734	5/338	13/330	6/401	41/366	1/41	4/38
**Positivity of doubtful cases (*n*, %)**		6 (50)	15 (25.9)	3 (60)	4 (30.8)	3 (50)	9 (21.9)	0 (0)	2 (50)
	Sensitivity (%, 95% CI)	90.6 (84.2–95.1)	95.8 (90.5–98.6)	94.1 (80.3–99.3)	93.9 (79.8–99.3)	87.7 (78.5–93.9)	96 (88.8–99.2)	100 (75.3–100)	100 (71.5–100)
	Specificity (%, 95% CI)	98.0 (96.6–98.9)	97.6 (96.0–98.6)	100 (98.8–100)	99.7 (98.1–100)	95.9 (93.2–97.8)	95.5 (92.5–97.6)	100 (87.7–100)	96.3 (81–99.9)
	PPN (%, 95% CI)	89.9 (83.4–94.5)	88.4 (81.5–93.3)	100 (89.1–100)	96.9 (83.8–99.9)	84.5 (75–91.5)	84.7 (75.3–91.6)	100 (75.3–100)	91.7 (61.5–99.8)
	NPV (%, 95% CI)	98.2 (96.8–99.0)	99.2 (98.1–99.7)	99.3 (97.7–99.9)	99.3 (97.6–99.9)	96.8 (94.3–98.5)	98.9 (96.9–99.8)	100 (87.7–100)	100 (86.8–100)
	AUC (95% CI)	0.943 (0.917–0.969)	0.967 (0.948–0.986)	0.971 (0.93–1.000)	0.968 (0.927–1.000)	0.918 (0.88–0.956)	0.958 (0.932–0.983)	1 (1–1)	0.981 (0.945–1.000)

* Doubtful cases are excluded in the sensitivity, specificity, PPV, NPV, and ROC area calculation. CI: 95% confidence interval; NA: not applicable; NC: not calculable.

**Table 4 diagnostics-15-00491-t004:** Acute joint dislocation.

		Overall Cases		Large Joints		Small Joints		Flat/Long Bones	
		(*n* = 792)		(*n* = 343)		(*n* = 407)		(*n* = 42)	
		Radiology Resident	IA Software	Radiology Resident	IA Software	Radiology Resident	IA Software	Radiology Resident	IA Software
**Acute joint dislocation (ratio of doubtful cases */certain)**		3/789	2/790	2/341	2/341	0/407	0/407	1/41	0/42
**Positivity of doubtful cases (*n*, %)**		2 (66.7)	0 (0)	1 (50)	0 (0)	0 (0)	0 (0)	1 (100)	0 (0)
	Sensitivity (%, 95% CI)	77.8 (52.4–93.6)	35.0 (15.4–59.2)	84.6 (54.6–98.1)	35.7 (12.8–64.9)	66.7 (9.43–99.2)	66.7 (9.43–99.2)	50 (1.26–98.7)	NC
	Specificity (%, 95% CI)	100 (99.5–100)	99.7 (99.1–100)	100 (98.9–100)	100 (98.9–100)	100 (99.1–100)	99.5 (98.2–99.9)	100 (91–100)	NC
	PPN (%, 95% CI)	100 (76.8–100)	77.8 (40–7.2)	100 (71.5–100)	100 (47.8–100)	100 (15.8–100)	50 (6.76–93.2)	100 (2.5–100)	NC
	NPV (%, 95% CI)	99.5 (98.7–99.9)	98.3 (97.2–99.1)	99.4 (97.8–99.9)	97.3 (95–98.8)	99.8 (98.6–100)	99.8 (98.6–100)	97.5 (86.8–99.9)	NC
	AUC (95% CI)	0.889 (0.79–0.988)	0.674 (0.566–0.781	0.923 (0.821–1)	0.679 (0.548–0.809)	0.833 (0.507–1)	0.831 (0.504–1)	0.75 (0.26–1)	NC

* Doubtful cases are excluded in the sensitivity, specificity, PPV, NPV, and ROC area calculation. CI: 95% confidence interval; NA: not applicable; NC: not calculable.

**Table 5 diagnostics-15-00491-t005:** Elbow joint effusion.

		Overall Cases	
		(*n* = 792)	
		Radiology Resident	IA Software
**Elbow joint effusion (ratio of doubtful cases */certain)**		0/28	4/24
**Positivity of doubtful cases (*n*, %)**		0 (0)	1 (25)
	Sensitivity (%, 95% CI)	100 (59–100)	100 (54,1–100)
	Specificity (%, 95% CI)	90.5 (69.6–98.8)	94.4 (72.7–99.9)
	PPN (%, 95% CI)	77.8 (40–97.2)	85.7 (42.1–99.6)
	NPV (%, 95% CI)	100 (82.4–100)	100 (80.5–100)
	AUC (95% CI)	0.952 (0.888–1.000)	0.972 (0.918–1.000)

* Doubtful cases are excluded in the sensitivity, specificity, PPV, NPV, and ROC area calculation. CI: 95% confidence interval; NA: not applicable; NC: not calculable.

**Table 6 diagnostics-15-00491-t006:** “Other findings” that Milvue has not been trained to detect.

		Overall Cases		Large Joints		Small Joints		Flat/Long Bones	
		(*n* = 792)		(*n* = 343)		(*n* = 407)		(*n* = 42)	
		Radiology Resident	IA Software	Radiology Resident	IA Software	Radiology Resident	IA Software	Radiology Resident	IA Software
Chronic fracture (ratio of doubtful cases */certain)		4/788	NA	2/341	NA	2/405	NA	0/42	NA
Positivity of doubtful cases (*n*, %)		1 (25)		1 (50)		0 (0)		0 (0)	
	Sensitivity (%, 95% CI)	29.2 (12.6–51.1)		12.5 (0.316–52.7)		35.7 (12.8–64.9)		50 (1.26–98.7)	
	Specificity (%, 95% CI)	99.9 (99.3–100)		100 (98.9–100)		99.7 (98.6–100)		100 (91.2–100)	
	PPN (%, 95% CI)	87.5 (47.3–99.7)		100 (2.5–100)		83.3 (35.9–99.6)		100 (2.5–100)	
	NPV (%, 95% CI)	97.8 (96.5–98.7)		97.9 (95.8–99.2)		97.7 (95.8–99)		97.6 (87.1–99.9)	
	AUC (95% CI)	0.645 (0.552–0.738)		0.563 (0.44–0.685)		0.677 (0.547–0.808)		0.75 (0.26–1.0)	
Arthropathy (ratio of doubtful cases */certain)		0/792	NA	0/343	NA	0/407	NA	0/42	NA
Positivity of doubtful cases (*n*, %)		0 (0)		0 (0)		0 (0)		0 (0)	
	Sensitivity (%, 95% CI)	70.1 (62.2–77.1)		79.8 (70.5–87.2)		51 (36.3–65.6)		66.7 (29.9–92.5)	
	Specificity (%, 95% CI)	97.6 (96.1–98.7)		98.4 (95.9–99.6)		96.9 (94.6–98.5)		100 (89.4–100)	
	PPN (%, 95% CI)	88 (81–93.1)		95.2 (88.1–98.7)		69.4 (51.9–83.7)		100 (54.1–100)	
	NPV (%, 95% CI)	93 (90.7–94.8)		92.3 (88.4–95.2)		93.5 (90.5–95.8)		91.7 (77.5–98.2)	
	AUC (95% CI)	0.839 (0.802–0.875)		0.891 (0.85–0.931)		0.74 (0.668–0.811)		0.833 (0.67–0.997)	
Focal lesion (ratio of doubtful cases */certain)		1/791	NA	1/342	NA	0/407	NA	0/42	NA
Positivity of doubtful cases (*n*, %)		0 (0)		0 (0)		0 (0)		0 (0)	
	Sensitivity (%, 95% CI)	6.67 (0.169–31.9)		NA		20 (0.505–71.6)		NA	
	Specificity (%, 95% CI)	99.4 (98.5–99.8)		NA		98.8 (97.1–99.6)		NA	
	PPN (%, 95% CI)	16.7 (0.421–64.1)		NA		16.7 (0.421–64.1)		NA	
	NPV (%, 95% CI)	98.2 (97–99)		NA		99 (97.5–99.7)		NA	
	AUC (95% CI)	0.53 (0.465–0.596)		NA		0.594 (0.398–0.79)		NA	
Anatomical variant (ratio of doubtful cases */certain)		0/792	NA	0/342	NA	0/407	NA	0/42	NA
Positivity of doubtful cases (*n*, %)		0 (0)		0 (0)		0 (0)		0 (0)	
	Sensitivity (%, 95% CI	64 (53.8–73.4)		90 (76.3–97.2)		46.7 (33.7–60)		NA	
	Specificity (%, 95% CI)	98.6 (97.4–99.3)		99 (97.1–99.8)		98 (95.9–99.2)		NA	
	PPN (%, 95% CI)	86.5 (76.5–93.3)		92.3 (79.1–98.4)		80 (63.1–91.6)		NA	
	NPV (%, 95% CI)	95 (93.1–96.5)		98.7 (96.7–99.6)		91.4 (88.1–94)		NA	
	AUC (95% CI)	0.813 (0.765–0.86)		0.945 (0.898–0.992)		0.723 (0.659–0.787)		NA	

* Doubtful cases are excluded in the sensitivity, specificity, PPV, NPV, and ROC area calculation. CI: 95% confidence interval; NA: not applicable; NC: not calculable.

## Data Availability

The original data presented in the study are openly available in FigShare at doi:10.6084/m9.figshare.28250348.

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
