# Peer review of "Diagnostic Performance of an Artificial Intelligence Software for the Evaluation of Bone X-Ray Examinations Referred from the Emergency Department"

_diagnostics, 2025, doi:10.3390/diagnostics15040491_

Round 1
Reviewer 1 Report
Comments and Suggestions for Authors
In order to assess an AI tool's potential as a trustworthy screening tool, this study compares its performance to that of radiologists and radiology residents. The AI tool is intended to identify fractures, joint dislocations, and elbow effusions in bone X-rays from the emergency room.
1) Can the authors include data from different centers and a variety of patient groups to make the study more applicable to real-world settings?
2) To make the paper better, the authors could explain why the AI struggled to detect acromioclavicular dislocations and suggest ways to improve its training.
3) Compare the AI's performance with emergency doctors to show how it works in real-life emergency situations.
4) Add ideas on how AI can fit into workflows without making extra work for radiologists.
5) The authors should examine how doubtful cases affect diagnoses and suggest ways to reduce uncertainty in clinical practice.
Author Response
Comments 1: Can the authors include data from different centers and a variety of patient groups to make the study more applicable to real-world settings?
Response 1: As mentioned in lines 135-138, our research team was not involved in the development or programming of the AI software. Our role was solely as end users, acting as independent testers of a commercially available product, without any direct commercial relationship with the company. However, further research is needed to study larger samples that more reliably represent real-life conditions.
Comments 2: To make the paper better, the authors could explain why the AI struggled to detect acromioclavicular dislocations and suggest ways to improve its training.
Response 2:
As we added in lines 457-460, this leads us to propose that future updates to the AI system should focus on expanding the training dataset to include a larger sample of acromioclavicular joint cases, particularly those with pathological conditions, in order to improve the system’s ability to accurately detect these dislocations. Additionally, in lines 461-466, we mention the feedback our team provided to the AI company.
Comments 3: Compare the AI's performance with emergency doctors to show how it works in real-life emergency situations.
Response 3: As we added in lines 384-390, in our study, we aim to reflect routine clinical practice, not only evaluating the accuracy of Milvue but also comparing the AI results with those of a radiology resident. This comparison is important, as the interpretation of a trauma specialist can differ significantly from that of emergency department clinicians, which may complicate direct comparisons. To make the evaluation more relevant, we chose to use the radiology resident's readings as a closer approximation to the interpretations of emergency department clinicians.
Comments 4: Add ideas on how AI can fit into workflows without making extra work for radiologists.
Response 4:
As we added in lines 493-505 in the discussion section, after analyzing our study results and the literature, we conclude that both the AI system and the radiology resident showed NPVs above 95% and AUC values above 0.8, except for joint dislocations, likely due to their low prevalence. These results, along with high 95% confidence intervals, suggest that AI can be an effective screening tool in emergency departments, identifying normal musculoskeletal radiographs and easing radiologists' workload. However, since many doubtful results were not confirmed as true positives, emergency physicians should continue consulting radiologists. While AI can assist in diagnostics, it cannot replace the human expertise and judgment required in complex cases.
Comments 5: The authors should examine how doubtful cases affect diagnoses and suggest ways to reduce uncertainty in clinical practice.
Response 5: As we added in lines 501-505, many doubtful results were not confirmed as true positives, and emergency physicians should continue consulting radiologists. While AI can assist in diagnostics, it cannot replace the human expertise and judgment required in complex cases.
Reviewer 2 Report
Comments and Suggestions for Authors
The article is devoted to the evaluation of the diagnostic effectiveness of artificial intelligence in the analysis of bone X-rays, which is an urgent topic in modern radiology. Using machine learning technologies to automatically detect fractures, dislocations, and other pathologies can significantly improve diagnosis and speed up the treatment process, especially in emergency settings. Thus, the relevance of the work does not raise any questions.
The authors have worked out the analysis of the subject area in sufficient detail, giving a large number of references to existing research both at the beginning of the work and during its discussion.
The experimental part is quite detailed and includes a large volume of tables with experimental results, as well as visual graphic material showing how machine learning recognizes fractures on bone X-rays.
The methodology section, as it seemed to me, is the weakest part of the presented work. The authors described the equipment used, briefly talked about the software and the data collection process. I would like to see a more formally presented research methodology in order to evaluate the authors' scientific contribution to the subject area. As part of this general comment, I would like to mention the following points:
1. Lines 115-120 mention that the deep learning algorithm is trained on 1.2 million and 157 thousand images. Do these datasets have designations? Has this process been considered in previous studies?
2. Section 2.3 also discusses machine learning-based software. I understand that this software was created by the authors' organization? It's worth giving more links to it and talking about it in a little more detail. If it was not considered in the author's works, then I would like to see the architecture of the machine learning models that they used.
3. Remove the red underscores in Figure 1.
Thus, in general, the level of the article is quite high, but I would like to see improvements in the direction I have indicated.
Author Response
Comments 1: 1. Lines 115-120 mention that the deep learning algorithm is trained on 1.2 million and 157 thousand images. Do these datasets have designations? Has this process been considered in previous studies?
Response 1:
We have added these clarifications in the Materials and Methods section, lines 129-138: According to information provided by the commercial company, the deep learning algorithm within the AI musculoskeletal product consists of a convolutional neural network that detected the aforementioned findings in each radiographic view. The model was trained and validated using 1,262,467 and 157,181 radiographic images. Afterwards, it was calibrated on another independent database of 4,759 images. The data used for model development were sourced from a multicenter database, which included both pediatric and adult patients. However, our research team was not involved in the development or programming of the AI software. Our role was solely as end users, acting as independent testers of a commercially available product, without any direct commercial relationship with the company.
Comments 2: Section 2.3 also discusses machine learning-based software. I understand that this software was created by the authors' organization? It's worth giving more links to it and talking about it in a little more detail. If it was not considered in the author's works, then I would like to see the architecture of the machine learning models that they used.
Response 2:
We have added these clarifications in the Materials and Methods section, lines 129-138: According to information provided by the commercial company, the deep learning algorithm within the AI musculoskeletal product consists of a convolutional neural network that detected the aforementioned findings in each radiographic view. The model was trained and validated using 1,262,467 and 157,181 radiographic images. Afterwards, it was calibrated on another independent database of 4,759 images. The data used for model development were sourced from a multicenter database, which included both pediatric and adult patients. However, our research team was not involved in the development or programming of the AI software. Our role was solely as end users, acting as independent testers of a commercially available product, without any direct commercial relationship with the company.
Comments 3: Remove the red underscores in Figure 1.
Response 3: We have corrected the underlining in Figure 1, which in the new version of the manuscript is now Figure 2.
Reviewer 3 Report
Comments and Suggestions for Authors
The manuscript evaluates the sensitivity, specificity and concordance of AI software in detecting fractures, joint dislocations, and elbow effusions. The study uses a retrospective design with 792 cases and compares AI performance to radiology residents and senior radiologists. It is well structured and well written. However, there are some points need to be addressed:
1. Please explain on why the selected AI software is particularly suited for emergency settings.
2. Include a flowchart to improve methodological clarity.
3. Provide a more in-depth analysis of the reasons behind the AI's lower sensitivity for detecting dislocations.
Highlight in the discussion the potential implications of the findings for clinical practice.
Author Response
Comments 1: Please explain on why the selected AI software is particularly suited for emergency settings.
Response 1:
We have addressed this issue in lines 493-501: After analyzing the results of our study and those found in the literature, we can conclude that both the AI system and the resident demonstrated NPVs greater than 95% and AUC values above 0.8, with the exception of joint dislocations, likely due to their low prevalence in our sample. These results were accompanied by high 95% CIs, indicating statistically reliable performance. This suggests that AI has the potential to serve as an effective screening tool in emergency departments, efficiently identifying normal musculoskeletal radiographs and integrating smoothly into the workflow. By enhancing clinicians' autonomy, AI can help reduce the workload of radiologists, allowing them to focus on more complex cases.
Comments 2: Include a flowchart to improve methodological clarity.
Response 2: We have added this in Figure 1, where we mention that after applying inclusion and exclusion criteria, 792 patients with X-rays of the appendicular skeleton and pelvis were included in the final sample. Musculoskeletal radiographs uploaded to the PACS were automatically analyzed by AI, which generates a separate report to be displayed alongside the original X-ray so that both radiologists and clinicians can review it.
Comments 3: Provide a more in-depth analysis of the reasons behind the AI's lower sensitivity for detecting dislocations.
Response 3: As mentioned in the discussion section, the prevalence of dislocations in the study sample analyzed by the AI program was very low. We recommend that future studies focus specifically on this type of pathology, using a larger sample of pathological cases, in order to obtain more consistent statistical results
Comments 4: Highlight in the discussion the potential implications of the findings for clinical practice.
Response 4: As we added in lines 493-505 in the discussion section, after analyzing our study results and the literature, we conclude that both the AI system and the radiology resident showed NPVs above 95% and AUC values above 0.8, except for joint dislocations, likely due to their low prevalence. These results, along with high 95% confidence intervals, suggest that AI can be an effective screening tool in emergency departments, identifying normal musculoskeletal radiographs and easing radiologists' workload.
Round 2
Reviewer 2 Report
Comments and Suggestions for Authors
I have studied the revised version of the article. The graphic material has been improved and expanded. Taking into account the answers of the authors, the idea of the end study became clear to me: checking the operability of an AI-based diagnostic tool in real conditions. This goal was achieved by the authors, and the AI was evaluated in comparison with human experience at different levels. Taking into account the above, the work can be accepted for publication.